# Can Direct Marketing Increase Fishery Profitability and Environmental Quality? Empirical Evidence of Aquaculture Farm Households in Taiwan

**Tzong-Haw Lee [1], Song-Yue Liu [2], Chiou-Lien Huang [3], Hung-Hao Chang [4] and Jiun-Hao Wang [5,*]**

1   School of Economics and Management, Hubei Polytechnic University, Guilin 435003, China; tzonghawlee@hbpu.edu.cn
2   School of Management, Wuhan University of Technology, Wuhan 430070, China; 257726@whut.edu.cn
3   Department of Future Studies and LOHAS Industry, Fo Guang University, Yilan 262307, Taiwan; chlihuang@mail.fgu.edu.tw
4   Department of Agricultural Economics, National Taiwan University, Taipei 10617, Taiwan; hunghaochang@ntu.edu.tw
5   Department of Bio-Industry Communication and Development, National Taiwan University, Taipei 10617, Taiwan
*   Correspondence: wangjh@ntu.edu.tw

**Abstract:** Marketing strategies play a significant role in determining farm income. Although direct marketing has been proposed as an innovative way to improve producers' economic welfare, little is known about producers' adoption of direct marketing among aquaculture farms. This study examines the adoption of wholesaler markets, individual wholesalers or shippers, and direct marketing among aquaculture farms. In addition, we quantify the effects of the use of different marketing channels on fishery revenues, profits, and production inputs. A sample of 25,180 aquaculture family farms in Taiwan was drawn from the fishery census survey. After estimating the simultaneous equation system model, we find that the use of multiple marketing channels generates the highest fishery revenues, which highlights the importance of marketing channel diversity on selling fishery products. Moreover, we find a positive effect of direct marketing on fishery revenues and profits. We also find that the use of direct marketing can reduce the use of groundwater in aquaculture production. Since the decrease in groundwater use can mitigate the severity of land subsidence, this paper provides evidence that direct marketing can possibly provide a win-win strategy to improve fishery producers' revenues and environmental quality.

**Keywords:** direct marketing; marketing channels; aquaculture production; land subsidence; groundwater use

## 1. Introduction

Marketing strategies are crucial for farms to sell their products, and they play a significant role in determining farm income. Small-scale family farms often rely heavily on traditional marketing channels such as wholesaler markets or individual distributors to sell their farm products [1]. While using traditional marketing channels may help family farms to manage market price uncertainty, the revenues generated from the final product sales are shared with the middlemen, potentially resulting in unequal distribution of the profits [2]. Direct marketing, also known as direct-sales-to-consumers marketing, has gained significant attention in recent years, and it has been considered an innovative marketing strategy to increase farm producers' revenues [3]. A considerable body of literature has been focused on direct-marketing strategies for crop farms, but not much research attention has been paid to fishery farms. An interesting question is whether the existing evidence found among crop farms in direct-marketing studies can be directly applied to aquaculture farm households. However, the answer is likely negative, as there

are significant differences between these two types of farms. For instance, fishery products have higher market values than crops, resulting in aquaculture farm households having less participation in non-farm labor markets compared to crop farm households. These differences in reliance on non-farm income sources may result in different tendencies to adopt marketing strategies. Furthermore, the nature of farm products is quite different when comparing crop and fishery farms. Consumers prefer fresh fishery products, making direct marketing arrangements a promising strategy for fishers to increase revenues by positioning themselves as both producers and dealers [3].

This paper contributes to the growing research interest in direct marketing by providing a quantitative analysis of aquaculture farm households' choice of marketing channels and producers' business performance using Taiwan as a case study. The objective of this study is to provide answers to the following questions: First, what is the role of the farm operator's socio-demographic characteristics, household conditions and production practices on the adoption of different marketing strategies of fishery farms? We focus on three marketing channels: the sale to wholesalers or distributors, those of wholesale markets, and those directly to consumers or restaurants. Second, what are the effects of the marketing channels on fishery farms' revenues, profits, and inputs used in fish production? Finally, since it has been well documented that aquaculture production has resulted in severe land subsidence problems by pumping groundwater during production in Taiwan, we examined the externality of producers' use of different marketing channels on the use of groundwater in aquaculture production. This part of the analysis sheds light on the relationship between marketing strategy and environmental sustainability because aquaculture production has resulted in serious pollution and land subsidence [4,5]. Due to the scarcity of fresh water in Taiwan, the over-pumping of groundwater in aquaculture production is common, making Taiwan a good study area for addressing this topic [6].

In Taiwan, there are 39,914 hectares of land used in aquaculture production. The number of family-type fish producers accounts for a large proportion of the fish producer population in Taiwan. The majority of fishery farms in Taiwan rely on traditional marketing channels by selling their products to individual wholesalers or wholesaler markets. The low amount of fish products makes individual wholesalers the major marketing channel for small fishery farms. In contrast, farms producing relatively larger amounts of fishery products are more likely to sell their products to wholesaler markets as they can have stronger bargaining power during price negotiation and lower transportation costs compared to small fishery farms [7]. In the past two decades, Taiwan's government encouraged fishery farms to use direct marketing channels by sell their products directly to individual consumers, restaurants or supermarkets. This active action on the promotion of direct marketing channels constitutes the response to the increased consumer concern for food safety. Direct marketing comes with the promotion of certificates or labels to ensure food safety in fishery production practices. For example, the Hazard Analysis Critical Control Point (HACCP) procedure has been implemented in Taiwan since 2004. The certificate of the HACCP requires producers to disclose detailed information, such as the use of water resources and medicine in fishery production [8]. Another popular food safety label in Taiwan is the Traceability Agricultural Product (TAP), which was implemented in Taiwan in 2007. The TAP system documents information that includes the name of the producers, the use of inputs in production, etc. [9]. Consumers can then use the QR code to trace the product records. Although consumers or restaurants are willing to pay a higher price to purchase fish products with food labels, the requirements of the food labels procedure generate additional costs to fishery farms [10].

The remaining part of this paper is organized as follows. We introduce the data and analytical framework in the next section. In what follows, we present the results and offer a discussion and the policy implications of these findings. In the final section, we conclude this paper and present the potential limitations of this paper and a direction for future studies.

## 2. Materials and Method

### 2.1. Data

To monitor the fishery industry in Taiwan, the Directorate-General of Budget, Accounting, and Statistics in Taiwan conducted face-to-face in-person interviews with all of the registered fish producers every five year since 1970. Given that the number of fishery farm households accounts for almost 98% of fish producers in Taiwan, this survey is a population-representative dataset of fish-producing households. In the survey, one principal operator in charge of fish production practices and business operation in each case is identified. The principle operator is responsible for reporting details on fish production and family characteristics. Information on fish production includes the revenues or sales value of fish products and the use of production inputs. Information on the socio-demographic characteristics of the principal operator, including gender, age, education and time allocation between self-family fish production and non-fishery work, was documented. In this study, we use the census survey conducted in 2015 [11] based on two reasons. First, the 2015 dataset is the latest fishery census survey. Second, this survey includes not only the standard questions as documented in early waves, but also several unique questions regarding the major marketing channels used by each fishery farm household to sell their products. So far, the 2015 census survey is the only data source that documents the use of marketing channels among fishery farms in Taiwan.

The 2015 census survey includes 38,800 fishery farm households. Each fishery farm household was asked to identify its main type of fishery production in 2015. In the survey, seven different types of fisheries are identified: the far sea fishery, offshore fishery, coastal fishery, inland fishery, marine aquaculture, inland brackish water aquaculture, and freshwater aquaculture. The last three categories belong to aquaculture production. Since this study focuses on the interaction between the use of marketing channels and fish production practices, especially the use of groundwater, we limit our sample to aquaculture farms, which includes the type of marine, inland brackish water, and freshwater aquaculture. In total, our sample consists of 25,192 aquaculture farm households. After subsequently deleting observations with missing values, our final sample included 25,180 aquaculture family farm households.

With respect to marketing channels, a survey question with multiple choices is documented. Each farm household was asked to select whether it engaged in the following marketing channels to sell their fishery products: wholesaler markets, fishery groups, individual shippers or wholesalers, supermarkets or hypermarkets, retailers, processing factories, restaurants, and individual consumers. According to the nature of the different marketing channels, we categorized these choices into three types of marketing channels. The first type is the wholesaler market, which included farm households that sold fishery products to wholesaler markets or fishery groups. The second type of marketing channel included those that sold fishery products to individual wholesalers, shippers or distributors. The third type of marketing channel included farms that sold products to supermarkets or hypermarkets, restaurants, or individual consumers. In the survey, each fishery farm household could select more than one type of marketing channel.

With respect to the economic performance of the aquaculture farm household, we specified a continuous variable to measure the sales value of self-produced fish products and other activities such as revenue from processing of self-produced fishery products and fishery tourism. Due to the limitation of the survey, revenue or income from non-fishery work is not included. The second variable is the profit of fish production, which is defined as the fishery revenue minus the expense of inputs used in fish production. These two variables are self-reported by the principle farm operator in each household, and both of them are measured in New Taiwan Dollars (NTD). Information on inputs used in fish production is also documented in the survey. We defined one continuous variable for the size of land used in aquaculture production and another continuous variable to indicate the number of hired labor used in aquaculture production. Although we did not have information on labor use and revenue from non-fishery work, the survey

documented whether the principle farm operator engaged in non-fishery work in the survey year. Accordingly, we defined a dummy variable to capture the extensive margin of the aquaculture farm in the non-fishery labor market. It has been well documented that aquaculture production is associated with the severe problem of land subsidence in Taiwan [4,6,12]. Since one research objective in this paper is to examine whether the use of marketing channels has any impact on environmental quality, we defined a dummy variable to indicate if the fishery farm used groundwater as the major water source in its aquaculture production.

We also specified several categories of explanatory variables associated with the choice of marketing channels and household income. For the demographic characteristics of the operator of the fishermen household, we defined five dummy variables to indicate if the operator was less than 29, 30–39, 40–49, 50–59, 60–69 or above 70 years of age. Five variables were specified to capture the operator's education level: illiterate, finished elementary school, junior high, senior high, and college or higher education. A dummy variable was also specified for the gender of the operator.

Several dummy variables are specified to indicate the major type of fish species. We defined a series of dummy variables by species in aquaculture cultivation: grouper, milkfish, tilapia, shrimp, oyster, clam, and other types of species. To control for family structure, we created three variables to measure household size and the ratio of adults living in the household. Two continuous variables were specified for the number of male and female family members, respectively. We also defined a variable to indicate the share of the number of adult members living in the family.

Since the use of marketing channels is assumed to be highly associated with the environmental condition of the fish markets, and since one of the marketing channels in this study is the wholesaler market, we defined three variables to capture the wholesaler market condition in the county in which each fishery farm is located. These include the average number of employees, land area, and monetary investments in equipment in the wholesale market. These variables are drawn from the Statistics Yearbook of fishery production in 2015 [13].

To understand the relationship between the engagement in marketing channels and the economic performance and production practice of aquaculture farms, we report the sample means of fishery revenue, profit, non-fishery work, number of hired labor, size of land in aquaculture production, and the use of groundwater in each combination of the three marketing channels in Table 1. For each outcome variable, we conducted an ANOVA test to see whether the sample means among the eight groups were statistically equal. As reported at the bottom of Table 1, the value of the F-tests ranged between 18.85 and 243.67. All of them reject the null hypothesis; this provides evidence that aquaculture farms using different marketing channels have different economic performances and inputs used in their fish production.

**Table 1.** Sample means of the outcome variables by marketing channels.

| N | Wholesale Markets | Wholesalers | Direct Marketing | Revenue (TWD Million) | Profit (TWD Million) | Non-Fishery Work (0/1) | Hired Labor (Person) | Land (Hectare) | Ground-water (0/1) |
|---|---|---|---|---|---|---|---|---|---|
| 1784 | No | No | No | 2.952 | 1.067 | 0.497 | 1.960 | 0.807 | 0.292 |
| 653 | No | No | Yes | 7.936 | 3.374 | 0.250 | 4.510 | 0.945 | 0.271 |
| 18,677 | No | Yes | No | 16.290 | 5.971 | 0.136 | 9.478 | 1.581 | 0.251 |
| 1631 | No | Yes | Yes | 14.129 | 5.249 | 0.142 | 9.641 | 1.828 | 0.149 |
| 550 | Yes | No | No | 16.476 | 6.529 | 0.229 | 7.480 | 1.603 | 0.275 |
| 158 | Yes | No | Yes | 14.519 | 5.473 | 0.196 | 6.703 | 1.691 | 0.222 |
| 1408 | Yes | Yes | No | 23.741 | 8.047 | 0.145 | 12.830 | 1.986 | 0.271 |
| 319 | Yes | Yes | Yes | 34.769 | 11.978 | 0.110 | 16.539 | 4.754 | 0.147 |
| F-test | | | | 77.53 | 67.20 | 243.67 | 82.30 | 46.54 | 18.85 |

Note: The total sample size is 25,180. The null hypothesis of the F-test is the equality of the sample mean across groups. The *p*-values of all the tests are smaller than 0.001.

Among the 25,180 aquaculture farm households, we found that 18,677 aquaculture farms sell fish products to wholesalers or shippers only (74%), with 158 aquaculture farms that engage in wholesaler markets and direct marketing simultaneously, which has the lowest ratio of marketing channel engagement. Aquaculture farms that use all three channels have the highest value of fishery revenue (TWD 34.769 million) and profit (TWD 11.978 million). This group of farms also has a higher value regarding the number of hired labor and the size of land used in aquaculture production. In contrast, the participation rates in non-fishery work and groundwater use are lower than in other groups.

In Table 2, we report the definition and sample statistics of the outcome variables and all of the explanatory variables in the full sample, as well as the use of the three marketing channels. As reported in Table 2, it appears that the socio-demographical characteristics of the farm operator, fishery production condition and family characteristics differ among aquaculture farms that use different marketing channels. For example, the farm operators engaging in wholesaler markets have higher education levels. Compared to other groups of farms, 30% and 12.3% of the farm operators that engage in wholesaler markets had finished senior high school and college, respectively. With respect to fishery production condition, marine aquaculture farms are more likely to engage in direct marketing channels to sell their products compared to other groups of aquaculture farms. This observation shows that in order to identify the impact of the use of direct marketing on fishery revenue and profit, it is necessary to control for the differences in explanatory variables among aquaculture farms that used different marketing channels.

**Table 2.** Sample statistics of the selected variables.

| Variable | Definition | All | | Wholesale Markets | | Wholesalers | | Direct Marketing | |
| --- | --- | --- | --- | --- | --- | --- | --- | --- | --- |
| | | Mean | S.D | Mean | S.D | Mean | S.D | Mean | S.D |
| Wholesaler markets | If use wholesaler markets (=1). | 0.097 | 0.296 | 1.000 | 0.000 | 0.078 | 0.269 | 0.173 | 0.378 |
| Wholesalers | If use wholesalers (=1). | 0.875 | 0.331 | 0.709 | 0.454 | 1.000 | 0.000 | 0.706 | 0.456 |
| Direct marketing | If use direct marketing (=1). | 0.110 | 0.312 | 0.196 | 0.397 | 0.088 | 0.284 | 1.000 | 0.000 |
| Revenue | Fishery revenue (TWD million). | 15.632 | 32.092 | 22.947 | 48.164 | 16.873 | 31.943 | 15.071 | 35.822 |
| Profit | Fishery profit (TWD million). | 5.711 | 11.792 | 8.052 | 18.259 | 6.137 | 11.776 | 5.596 | 14.689 |
| Non-fishery work | If operator has a non-fishery job (=1). | 0.167 | 0.373 | 0.163 | 0.369 | 0.136 | 0.343 | 0.167 | 0.373 |
| Hired labor | Number of hired labor (person). | 9.043 | 16.173 | 11.710 | 21.018 | 9.807 | 16.563 | 9.057 | 18.565 |
| Groundwater | If groundwater is the main water source in fish production (=1). | 0.248 | 0.432 | 0.253 | 0.435 | 0.244 | 0.429 | 0.182 | 0.386 |
| Land | Land area in fish production (hectare). | 1.589 | 3.893 | 2.243 | 8.279 | 1.671 | 4.043 | 1.949 | 7.903 |
| Age_29 | If operator age ≤29 (=1). | 0.006 | 0.075 | 0.004 | 0.064 | 0.006 | 0.075 | 0.004 | 0.063 |
| Age_3039 | If operator age 30–39 (=1). | 0.042 | 0.201 | 0.036 | 0.187 | 0.043 | 0.203 | 0.032 | 0.175 |
| Age_4049 | If operator age 40–49 (=1). | 0.140 | 0.347 | 0.143 | 0.350 | 0.142 | 0.349 | 0.130 | 0.336 |
| Age_5059 | If operator age 50–59 (=1). | 0.281 | 0.450 | 0.311 | 0.463 | 0.281 | 0.449 | 0.294 | 0.456 |
| Age_6069 | If operator age 60–69 (=1). | 0.280 | 0.449 | 0.284 | 0.451 | 0.278 | 0.448 | 0.298 | 0.457 |
| Age_70 | If operator age ≥70 (=1). | 0.251 | 0.434 | 0.222 | 0.416 | 0.251 | 0.433 | 0.243 | 0.429 |
| Illiteracy | If operator is illiterate (=1). | 0.088 | 0.283 | 0.058 | 0.234 | 0.087 | 0.282 | 0.071 | 0.256 |
| Elementary | If finished elementary school (=1). | 0.313 | 0.464 | 0.269 | 0.444 | 0.316 | 0.465 | 0.339 | 0.473 |
| Junior high | If finished junior high school (=1). | 0.234 | 0.423 | 0.250 | 0.433 | 0.235 | 0.424 | 0.230 | 0.421 |
| Senior high | If finished senior high school (=1). | 0.265 | 0.441 | 0.300 | 0.458 | 0.264 | 0.441 | 0.256 | 0.437 |
| College | If college or higher education (=1). | 0.101 | 0.302 | 0.123 | 0.328 | 0.097 | 0.296 | 0.104 | 0.306 |
| Male | If male operator (=1). | 0.856 | 0.351 | 0.861 | 0.346 | 0.857 | 0.350 | 0.873 | 0.333 |
| HHSIZE_male | Male household members (person). | 1.806 | 1.112 | 1.858 | 1.107 | 1.796 | 1.105 | 1.840 | 1.121 |
| HHSIZE_female | Female household members (person). | 1.558 | 1.196 | 1.628 | 1.236 | 1.546 | 1.192 | 1.576 | 1.207 |
| Ratio_adult | Ratio of adult household members. | 0.946 | 0.133 | 0.947 | 0.131 | 0.947 | 0.133 | 0.944 | 0.134 |
| Type_marine | If marine aquaculture (=1). | 0.094 | 0.292 | 0.055 | 0.228 | 0.104 | 0.305 | 0.137 | 0.344 |

**Table 2.** *Cont.*

| Variable | Definition | All | | Wholesale Markets | | Wholesalers | | Direct Marketing | |
|---|---|---|---|---|---|---|---|---|---|
| | | **Mean** | **S.D** | **Mean** | **S.D** | **Mean** | **S.D** | **Mean** | **S.D** |
| Type_brackish water | If inland brackish water aquaculture (=1). | 0.543 | 0.498 | 0.531 | 0.499 | 0.556 | 0.497 | 0.362 | 0.481 |
| Type_fresh water | If inland freshwater aquaculture (=1). | 0.363 | 0.481 | 0.415 | 0.493 | 0.340 | 0.474 | 0.502 | 0.500 |
| Aqua_grouper | If grouper aquaculture (=1). | 0.079 | 0.270 | 0.115 | 0.320 | 0.082 | 0.274 | 0.078 | 0.269 |
| Aqua_milkfish | If milkfish aquaculture (=1). | 0.239 | 0.427 | 0.304 | 0.460 | 0.253 | 0.435 | 0.229 | 0.420 |
| Aqua_tilapia | If tilapia aquaculture (=1). | 0.163 | 0.370 | 0.184 | 0.387 | 0.134 | 0.341 | 0.175 | 0.380 |
| Aqua_shrip | If shrimp aquaculture (=1). | 0.128 | 0.334 | 0.129 | 0.335 | 0.134 | 0.341 | 0.108 | 0.311 |
| Aqua_oyster | If oyster aquaculture (=1). | 0.088 | 0.283 | 0.045 | 0.207 | 0.093 | 0.291 | 0.157 | 0.364 |
| Aqua_clam | If clam aquaculture (=1). | 0.136 | 0.343 | 0.043 | 0.203 | 0.145 | 0.352 | 0.053 | 0.225 |
| Aqua_other | If other types of fish (=1). | 0.168 | 0.373 | 0.181 | 0.385 | 0.159 | 0.366 | 0.198 | 0.399 |
| City | If located in a city area (=1). | 0.398 | 0.489 | 0.557 | 0.497 | 0.393 | 0.488 | 0.453 | 0.498 |
| Mkt_employee | Number of employees in wholesaler markets (person). | 54.447 | 24.878 | 59.788 | 25.307 | 55.243 | 24.700 | 48.175 | 20.673 |
| Mkt_land | Land area of wholesaler markets (hectare) | 2.565 | 1.633 | 2.752 | 1.724 | 2.618 | 1.631 | 2.121 | 1.223 |
| Mkt_equip | Investment in equipment (TWD 1000/m$^2$). | 3.758 | 5.542 | 5.362 | 6.970 | 3.653 | 4.801 | 4.790 | 8.629 |
| N | | 25,180 | | 2435 | | 22,035 | | 2761 | |

### 2.2. Econometric Model

Several econometric issues have to be addressed in the specification of the empirical model. First, the choices of marketing channels are made by the aquaculture farms, so the problem of endogeneity bias has to be considered. Endogeneity bias may arise if the decision to choose marketing channels and the fishery revenue or other outcomes are correlated due to unobserved common factors, such as the risk preference of the farmers. It is likely that aquaculture farmers who are more averse to risk may invest less in farm equipment. Therefore, this type of farmer may have lower revenue than the others. Since the risk attitude of the farmer is not observed by the researcher, potential endogeneity bias can occur. The second issue is related to the selection of the approach. The framework of our analysis lies in the treatment effect or program evaluation literature. In this strand of literature, how to deal with endogeneity bias is the core issue. Several methods including the propensity score matching, difference-in-differences, and regression discontinuity have become popular in program evaluation literature (for a review of each model, see [14,15]). In this study, we did not apply these methods as they are more appropriately applied to a case with a single treatment. Even though multiple treatment models have been proposed, some strict restrictions have been imposed in these models. For example, the multi-valued treatment effect model proposed in Cattaneo [16] extends the propensity score method to the case of multiple treatments, although all of the treatments have to be mutually exclusive. Moreover, the identification condition of this model relies on the selection-on-observables assumption, which cannot be empirically tested. To cope with endogeneity bias, we follow the traditional simultaneous system framework to specify a four-equation simultaneous equation system:

$$
\begin{aligned}
D_{1i}^* &= \alpha_1 + \beta_1' X_i + \gamma_1' Z_i + \varepsilon_{1i} \\
D_{2i}^* &= \alpha_2 + \beta_2' X_i + \gamma_2' Z_i + \varepsilon_{2i} \\
D_{3i}^* &= \alpha_3 + \beta_3' X_i + \gamma_3' Z_i + \varepsilon_{3i} \\
Y_i &= \alpha + \lambda_1 \times D_{1i} + \lambda_2 \times D_{2i} + \lambda_3 \times D_{3i} + \beta' X_i + \varepsilon_i \\
D_{ki} &= 1 \text{ if } D_{ki}^* > 0; \ D_{ki} = 0 \text{ if } D_{ki}^* \le 0; \ k = 1, 2, 3
\end{aligned}
\tag{1}
$$

where $D^*_{1i,}$ $D^*_{2i}$ and $D^*_{3i}$ are the unobserved latent variables for the use of wholesaler markets, individual wholesalers or shippers, and direct marketing of the $i$th aquaculture farm household, respectively. $D_{ki}$ is the observed binary choice variable of each decision

($k$ = 1, 2, 3). $Y_i$ is the outcome variable for fishery revenue, profit, or production inputs. The vector $X_i$ includes explanatory variables associated with the socio-demographic characteristics of the farm operator, family and production condition, and $Z_i$ includes the condition of the wholesaler markets in the county in which each fishery farm is located (see the full list of the variables in Table 2). $\alpha, \alpha_1, \alpha_2, \alpha_3, \beta, \beta_1, \beta_2, \beta_3, \gamma_1, \gamma_2, \gamma_3, \lambda_1, \lambda_2, \lambda_3$ are the parameters of interest. $\varepsilon_1, \varepsilon_2, \varepsilon_3, \varepsilon$ are random errors that follow a multivariate normal distribution with

means zero, and the variance–covariance matrix is given by $\sum = \begin{bmatrix} 1 & \rho_{12} & \rho_{13} & \rho_{14} \\ \rho_{12} & 1 & \rho_{23} & \rho_{24} \\ \rho_{13} & \rho_{23} & 1 & \rho_{34} \\ \rho_{14} & \rho_{24} & \rho_{34} & \sigma^2 \end{bmatrix}$,

where the correlation coefficient between any two choices ($\rho$) captures the joint nature of these decisions. These correlation coefficients capture the relationships among the four equations due to unobserved common factors. Therefore, testing whether these parameters are statistically close to zero provides justification for the potential endogeneity bias problem. The parameters $\lambda_1, \lambda_2, \lambda_3$ capture the effects of the use of each marketing channel on the outcome variable. In the empirical analysis, Equation (1) is jointly estimated using the conditional mixed process proposed in Roodman [17], which utilizes the Geweke, Hajivassiliou, and Keane (GHK) algorithm to consistently estimate the full model.

With respect to model identification, Equation (1) is theoretically identified by the recursive nature between the use of marketing channels and the outcome variable, and the parametric assumption of the error terms. A recursive structure is ensured by the fact that the choice of the marketing channels affects the outcome variable, and not vice versa. This justification of the one-way causal relationship has also been discussed in the theoretical framework in the previous section. Unlike the instrumental variable approach, it is not necessary to have any exclusion variables to identify the system of equations [18]. Nevertheless, it is generally considered good empirical practice to include some exclusion variables to increase the statistical power underlying the empirical estimation. In this paper, we use the three variables to reflect the capacity and size of the wholesaler markets at the county level as exclusion variables (the variables $Z_i$ in Equation (1)). These variables are assumed to be directly correlated with the likelihood of aquaculture farms to engage in the wholesaler markets in the local area.

## 3. Results

We report our results in several tables. Table 3 reports the marginal effects of the explanatory variables in the simultaneous equation system model for the choice of marketing channels. In Table 4, we report the impacts on farm revenues, profit, and inputs associated with aquaculture production from the use of different marketing channels. For the sake of presentation, we only report the estimated coefficients of the marketing channels. In Table 5, we report the results of the statistical tests in regard to model specification.

**Table 3.** Estimated marginal effects of the use of marketing channels.

| Variable | Wholesale Markets | | | Wholesalers | | | Direct Marketing | |
| --- | --- | --- | --- | --- | --- | --- | --- | --- |
| | Mar. Eff | S.E | | Mar. Eff | S.E | | Mar. Eff | S.E |
| Age_3039 | 0.009 | 0.028 | | −0.008 | 0.030 | | 0.006 | 0.031 |
| Age_4049 | 0.018 | 0.027 | | −0.016 | 0.028 | | 0.039 | 0.029 |
| Age_5059 | 0.025 | 0.027 | | −0.023 | 0.028 | | 0.051 | 0.029 |
| Age_6069 | 0.026 | 0.027 | | −0.027 | 0.028 | | 0.052 | 0.029 |
| Age_70 | 0.018 | 0.027 | | −0.029 | 0.029 | | 0.043 | 0.030 |
| Elementary | 0.001 | 0.008 | | 0.028 *** | 0.008 | | 0.010 | 0.008 |
| Junior high | 0.016 | 0.009 | | 0.018 ** | 0.009 | | −0.004 | 0.009 |
| Senior high | 0.011 | 0.009 | | 0.016 | 0.009 | | 0.001 | 0.009 |

**Table 3.** *Cont.*

| Variable | Wholesale Markets | | | | Wholesalers | | | | Direct Marketing | | | |
|---|---|---|---|---|---|---|---|---|---|---|---|---|
| | Mar. Eff | | | S.E | Mar. Eff | | | S.E | Mar. Eff | | | S.E |
| College | 0.008 | | | 0.010 | −0.014 | | | 0.010 | 0.011 | | | 0.005 |
| Male | 0.001 | | | 0.006 | 0.003 | | | 0.006 | 0.013 | ** | | 0.006 |
| HHSIZE_male | 0.003 | | | 0.002 | −0.004 | ** | | 0.002 | 0.001 | | | 0.002 |
| HHSIZE_female | 0.004 | ** | | 0.002 | −0.007 | *** | | 0.002 | 0.001 | | | 0.002 |
| Ratio_adult | 0.014 | | | 0.016 | 0.023 | | | 0.017 | −0.020 | | | 0.016 |
| Aqua_grouper | 0.014 | | | 0.008 | 0.061 | *** | | 0.009 | −0.043 | *** | | 0.009 |
| Aqua_milkfish | 0.004 | | | 0.006 | 0.100 | *** | | 0.007 | −0.038 | *** | | 0.007 |
| Aqua_tilapia | 0.009 | | | 0.007 | −0.061 | *** | | 0.006 | −0.012 | | | 0.007 |
| Aqua_shrip | −0.015 | ** | | 0.007 | 0.079 | *** | | 0.008 | −0.027 | *** | | 0.007 |
| Aqua_oyster | −0.050 | ** | | 0.023 | 0.144 | *** | | 0.022 | −0.044 | ** | | 0.019 |
| Aqua_clam | −0.057 | *** | | 0.009 | 0.105 | *** | | 0.008 | −0.117 | *** | | 0.009 |
| Type_marine | 0.010 | | | 0.022 | −0.045 | ** | | 0.021 | 0.116 | *** | | 0.019 |
| Type_brackish water | −0.001 | | | 0.005 | −0.004 | | | 0.005 | 0.043 | *** | | 0.005 |
| City | 0.094 | *** | | 0.006 | −0.089 | *** | | 0.006 | 0.096 | *** | | 0.006 |
| Mkt_employee | 0.003 | *** | | 0.000 | −0.001 | *** | | −0.000 | 0.001 | *** | | 0.000 |
| Mkt_land | 0.046 | *** | | 0.003 | −0.041 | *** | | −0.004 | −0.051 | *** | | 0.004 |
| Mkt_equip | 0.001 | ** | | 0.000 | −0.001 | *** | | −0.000 | 0.000 | | | 0.000 |

Note: *** and ** indicate significance at the 1% and 5% level.

**Table 4.** Estimated results of fishery revenues, profits, and input use in fishery production.

| Variable | Fishery Revenue | | | | | | Fishery Profit | | | | | | Land in Fishery Production | | | | | |
|---|---|---|---|---|---|---|---|---|---|---|---|---|---|---|---|---|---|---|
| | (A1) Coef. | | S.E | (A2) Coef. | | S.E | (B1) Coef. | | S.E | (B2) Coef. | | S.E | (C1) Coef. | | S.E | (C2) Coef. | | S.E |
| Wholesaler markets | 1.58 | *** | 0.515 | 1.29 | *** | 0.343 | 0.45 | *** | 0.108 | 0.39 | *** | 0.098 | 0.07 | *** | 0.008 | 0.27 | *** | 0.086 |
| Wholesalers | 1.32 | *** | 0.113 | 1.39 | *** | 0.118 | 0.46 | *** | 0.038 | 0.50 | *** | 0.040 | 0.08 | *** | 0.008 | 0.44 | *** | 0.084 |
| Direct marketing | 1.07 | | 0.519 | 0.68 | | 0.358 | 0.07 | | 0.040 | 0.03 | | 0.018 | 0.04 | *** | 0.008 | 0.33 | ** | 0.154 |
| Direct marketing × wholesaler markets | | | | 0.05 | | 0.029 | | | | 0.13 | ** | 0.051 | | | | 0.05 | | 0.031 |
| Direct marketing × wholesalers | | | | 0.04 | ** | 0.016 | | | | 0.21 | *** | 0.059 | | | | 0.11 | *** | 0.016 |
| Direct marketing × markets × wholesalers | | | | 0.09 | ** | 0.033 | | | | 0.32 | ** | 0.123 | | | | 0.09 | ** | 0.036 |

| Variable | Non-fishery work | | | | | | Hired labor in fishery production | | | | | | Groundwater in fishery production | | | | | |
|---|---|---|---|---|---|---|---|---|---|---|---|---|---|---|---|---|---|---|
| | (D1) | | | (D2) | | | (E1) | | | (E2) | | | (F1) | | | (F2) | | |
| Wholesaler market | −0.06 | *** | 0.008 | −0.01 | *** | 0.002 | 0.429 | *** | 0.035 | 1.25 | *** | 0.079 | 0.01 | | 0.013 | 0.00 | | 0.014 |
| Wholesalers | −0.02 | *** | 0.007 | −0.03 | *** | 0.002 | 0.102 | | 0.052 | 0.74 | | 0.391 | 0.02 | ** | 0.010 | 0.01 | | 0.011 |
| Direct marketing | 0.04 | *** | 0.007 | 0.02 | *** | 0.002 | 0.133 | *** | 0.033 | 1.12 | | 0.065 | −0.04 | *** | 0.007 | −0.04 | *** | 0.013 |
| Direct marketing × wholesaler markets | | | | −0.00 | | 0.003 | | | | −0.21 | | 0.107 | | | | 0.01 | ** | 0.002 |
| Direct marketing × wholesalers | | | | 0.02 | *** | 0.002 | | | | 0.18 | ** | 0.061 | | | | 0.00 | ** | 0.001 |
| Direct marketing × markets × wholesalers | | | | 0.01 | | 0.038 | | | | 0.12 | | 0.124 | | | | −0.00 | | 0.003 |

Note: All of the explanatory variables are included in each equation. The full list of the explanatory variables is found in Table 2. *** and ** indicate significance at the 1% and 5% level.

**Table 5.** Results of the LR tests on model specification.

| Outcome Equation | H$^0$: ρ = 0 [#1] | H$^0$: Z = 0 [#2] |
|---|---|---|
| Fishery revenue | 111 | 243 |
| Fishery profit | 120 | 252 |
| Land in fish production | 121 | 251 |
| Number of hired labor | 109 | 241 |
| Non-fishery work | 231 | 238 |
| Groundwater use | 641 | 287 |
| Critical value | $x^2(6, 0.01) = 16.8$ | $x^2(9, 0.01) = 21.67$ |

Note: We conducted LR tests in the model without the inclusion of the interaction terms among marketing channels. [#1] H$^0$: all of the correlation coefficients are zero. [#2] H$^0$: the coefficients of the three variables related to wholesaler markets in the local area are zero.

*3.1. The Determinants of the Choice of Marketing Channels*

In the main model, we estimate a simultaneous equation system model with fishery revenue as the outcome variable by using the conditional mixed process method. We report the full estimation results in Table A1 in the Appendix A and the results of the calculated marginal effects of the explanatory variables in Table 3. As reported in Table 3, fishery production practice, the socio-demographic characteristics of the principle operator, and household conditions are associated with the aquaculture farms' choices of marketing channels. With respect to the socio-demographic characteristics of the operator, it is evident that operator's education is an important factor regarding the choices of marketing channels. Operators with higher education levels are more likely to engage in direct-sales-to-consumers marketing channels. For example, operators with a college degree or higher education are more likely to sell products directly to consumers or restaurants by 1.1 percentage points compared to the reference group of the operators who are illiterate, all things being equal. The gender of the operator also matters in relation to the choice of marketing channels. The results show that male operators are more likely to sell their products directly to consumers or restaurants by 1.3% compared to their female operator counterparts. Moreover, we found that fish species are important when determining the choices of marketing channels of aquaculture farms. For example, compared to the reference group of farms harvesting other types of fish species, groper aquaculture farms are more likely to engage in wholesalers by 6.1%, *ceteris paribus*.

*3.2. The Impact of Marketing Channels on Economic Performance*

In addition to fishery revenue, we estimate the simultaneous equation system model for five other outcome variables, including fishery profits, number of hired labor, size of land in production, non-fishery work, and groundwater use. Table 4 reports the estimated coefficients of the three marketing channels for each outcome equation. For each outcome variable, we specify and estimate two slightly different models. In addition to the explanatory variables, the first model only includes the separate variables of each marketing channel, while the second model includes the additional three interaction terms of the three marketing channel variables. The inclusion of these interaction terms can help to test whether the use of multiple marketing channels affects the outcome variables, especially the fishery revenues and profits.

As reported in columns (A1) in which fishery revenue is specified as the outcome variable, the use of wholesaler markets, wholesalers, and direct marketing all contribute positively to fishery revenue. Other things being equal, aquaculture farms that use wholesaler markets, individual wholesalers, and direct marketing channels have higher fishery revenues by TWD 1.582, TWD 1.322, and TWD 1.075 million, respectively, compared to their non-user counterparts. By further including the interaction terms of the use of marketing channels, the results reported in column (A2) show that using multiple marketing channels can further increase fishery revenue. For example, aquaculture farms that rely only on direct marketing have higher revenues by TWD 1.075 million compared to their non-user

counterparts of direct marketing. However, farms engaging in both direct marketing and wholesalers markets have higher revenues by TWD 0.046 million compared to those that simply rely on direct marketing channels. We find a similar pattern of the results for fishery profits (see columns (B1) and (B2)).

### 3.3. The Impact of Marketing Channels on Inputs Used in Aquaculture Production

We report the effects of the use of marketing channels on land size, engagement in non-fishery work, the amount of hired labor and the use of groundwater in columns (B1)–(F2), respectively. The results show that the use of marketing channels also increases the use of land in aquaculture production. Other things being equal, the use of wholesaler markets, individual wholesalers and direct marketing increase the size of production land by 0.075, 0.082 and 0.045 hectares compared to their non-user counterparts, respectively. In addition, using multiple marketing channels to sell fishery products results in more land used in aquaculture production. The consistency of the results in land use and fishery revenues may reflect the fact that land is an essential input in aquaculture production, and the increase in the size of aquaculture production can generate higher fishery revenue. Similarly, we find a positive effect of marketing channel use on the number of hired workers in fish production. The results regarding the use of hired labor are interesting. As reported in columns (E1), the magnitude of the effects is smaller for aquaculture farms that sold their products only to individual wholesalers (the coefficient is 0.102), and the largest effect is found for those that sold products to wholesaler markets (the coefficient is 0.429). These results may reflect the nature of the shipping process in that fishery products sold to wholesaler markets usually require a significant amount of transportation-related labor. Aquaculture farms that sell their products to individual wholesalers or distributors are not responsible for product shipping; therefore, they use less labor compared to the group of farms that sell products to wholesaler markets or directly to consumers.

As discussed in the conceptual framework, we find that the use of marketing channels is significantly associated with the aquaculture farms' engagement in the non-fishery labor market. However, different effects are evident for the different use of marketing channels. Results reported in column (D1) show that farms that sold products to wholesaler markets and individual wholesalers are less likely to engage in non-fishery labor market by 6.3 and 2.2 percentage points, respectively, compared to their counterparts of non-users. In contrast, we find a positive effect of direct marketing on non-fishery work. Aquaculture farms that sell products directly to consumers or restaurants are more likely to work off the farm by 3.7 percentage points, all things being equal.

To link our analysis to environmental quality, we conducted an analysis to examine the effects of the choice of marketing channels on groundwater use. The results are reported in columns (F1) and (F2). The results point to a negative effect of the use of marketing channels on groundwater use for aquaculture farms that sold products to wholesalers or wholesaler markets. In contrast, as reported in column (F1), aquaculture farms that sold products directly to consumers or restaurants are less likely to use groundwater as the main water resource in aquaculture production by 3.7 percentage points compared to their non-user counterparts, all things being equal.

### 3.4. Results of the Statistical Tests Regarding Model Specification

We conducted two statistical tests to show the validity of the model specification. The first test is used to justify the specification of the simultaneous equation system. We conducted the LR test to check whether the correlation coefficients are jointly equal to zero (i.e., $\rho_{12} = \rho_{23} = \rho_{24} = \rho_{13} = \rho_{34} = \rho_{14} = 0$). The second LR test is used to check the statistical power of the exclusion variables. As indicated earlier, the model is theoretically identified, but the inclusion of the exclusion variables can increase the statistical power in regard to the empirical estimation. We conducted these tests for the six simultaneous equation systems.

As reported in Table 5, the test values of the LR test range between 109 and 641 under the null hypothesis that all of the correlation coefficients are zero. Given that the critical value is 16.8 ($x^2$(6, 0.01) = 16.8), we reject the null hypothesis for all six models; this result justifies the use of the simultaneous equation system. With respect to the exclusion variables, the results of the LR test are between 238 and 287; all of them are larger than the critical value ($x^2$(9, 0.01) = 21.67). These results provide statistical evidence regarding the validity of these exclusion variables.

## 4. Discussion

Several interesting findings are revealed in this study, and we offer discussions on them in this section. We found that fishery revenues or profits are higher for aquaculture farms that involve multiple marketing channels. This result echoes the findings of the agricultural marketing literature, which has pointed out that given the nature of highly perishable agricultural products, optimizing sales likely requires the flexibility of combining different marketing channels capable of accepting alternative sizes and types of products. Moreover, the use of multiple marketing channels can help to reduce the price risk and increase farm revenues [19]. The policy implications inferred from this finding are straightforward. From the view of fishery producers, using multiple marketing channels can help to spread the operational risks. Therefore, the government should provide assistance or subsidies to increase the use of multiple marketing channels among fishery producers. In contrast, simply promoting a single marketing channel is not the best way to increase fishery producers' revenue.

Although the Taiwanese government has promoted the use of direct marketing among fishery producers, the adoption rate is still low among aquaculture farms. Most of the fishery farms in Taiwan still rely on traditional marketing channels by selling their products to wholesalers or distributors, which may reflect the fact that wholesale channels typically have a better ability to move large quantities of produce quickly and usually at a lower transportation cost than through direct channels. In contrast, direct marketing often requires more customer interaction and time requirements from the producers. The lower participation rate of direct marketing may also reflect the strict regulation of the safety of fish products when fish products are directly sold to consumers or restaurants.

Previous studies have highlighted the significance of marketing channels for fishery producers' income [20–26]. For instance, Ahmed et al. [23] examined the impact of a government-funded project aimed at promoting direct marketing among prawn producers in Bangladesh. They found that the marketing chain of prawn products became shorter, with a significant reduction in intermediaries after the implementation of the program. Direct marketing can also enhance producers' revenues since it provides them with a reasonable and stable purchasing price. Gomez and Maynou [24] studied fish producers' attitudes toward direct sales and the certification of origin labeling scheme labels in Catalonia and the Balearic Islands. They found that setting minimum fair ex-vessel prices would reduce the negative perception of the fishers regarding price competition. Wetengere [25] examined the constraints to the marketing of farmed fish in inland Tanzania. The authors highlighted the importance of market engagement on fishers' profits, with fish products sold to middlemen and shipped to urban cities having higher producer prices on average. Geng [26] used survey data to study the determinants of aquatic farmers' participation in marketing channels in Jiangsu Province, China. The authors found that farmers' social networks could increase their participation in modern marketing channels such as direct marketing. In contrast to the previous studies, we find that using direct marketing alone cannot generate the largest profits of fishery farms. This may reflect the strict requirement of food safety on fishery products in Taiwan. In Taiwan, restaurants usually require certificates to ensure food safety, such as the label for Hazard Analysis Critical Control Point (HACCP). The HACCP reveals information on fish production, such as the use of water resource and medicine, to ensure the safety of fish products sold to consumers. Although consumer willingness to pay for fish products with HACCP labels is higher than those

without labels [10], the required information disclosure process in aquaculture production increases the entry barriers for farms to apply for HACCP.

With respect to the determinants of the aquaculture farms that engage in direct marketing, we find that the human capital of the operator as well as the fish production characteristics are important factors. Aquaculture farms whose operators have higher education are more likely to adopt a direct marketing strategy. This result is consistent with the findings of prior studies on technology adoption in agriculture, which point out that educated farm workers are more likely to adopt new technologies because they have a better ability to acquire information on the new technology [27,28]. Interestingly, we also find that aquaculture farms that use direct marketing channels are more likely to engage in non-fishery work. This finding may reflect the importance of social connection in non-fish job markets on the adoption of direct marketing. For example, the successful operation of direct marketing usually requires access to, or the search for, potential customers [26]. In our case, aquaculture farms that have off-fishery business work may have more opportunities to search for potential consumers or restaurants. In this regard, a positive correlation between off-farm work and the adoption of direct marketing is expected.

Our results also indicate that aquaculture farms engaging in direct marketing are less likely to use groundwater as the main water resource in their aquaculture production. This result again echoes the strict requirement of HACCP labels on fishery products sold to individual consumers due to the concern for safe food. Many aquaculture farms use groundwater in production; however, pumping groundwater is illegal in Taiwan since groundwater use is highly associated with the land subsidence problem. Therefore, those fishery farms that heavily rely on groundwater may encounter difficulty in receiving a food safety certificate. In terms of policy, our finding is important from the standpoint of policy since we provide supporting evidence that policies that aim to increase aquaculture farms' adoption of direct marketing have an unintended effect on environmental quality, such as land subsidence. More specifically, our results provide an interesting case study that direct marketing can prove to be a win-win strategy to secure fishery revenue; it also has the potential to improve environmental quality.

Finally, we summarize the contributions of our study to previous studies on marketing channels as follows. Firstly, unlike previous studies that focused solely on a single marketing channel, such as direct marketing, this study examined both direct marketing and traditional wholesale channels. As far as we know, this is one of the first papers to compare the effects of these two distinct marketing channels on fishery economic outcomes. Secondly, prior studies that explored the relationship between marketing channels and fishery income often used simplistic descriptive statistics that failed to account for endogeneity bias due to fishery farms' marketing channel choices. This study used econometric analysis to control for differences in the socio-demographic characteristics of the operator, household, and production conditions among various groups of aquaculture farms. As a result, it provides a more accurate assessment of the impacts of different marketing channels on fishery outcomes. Thirdly, the data used in this study are unique. They relied on a census survey of aquaculture farm households in Taiwan, which provides more objective policy implications. Finally, this study examined the interactions between aquaculture farms' marketing strategies and groundwater use. Since aquaculture production is highly associated with land subsidence caused by groundwater over-pumping, this analysis has implications for environmental sustainability by examining the impact of marketing channels on groundwater use.

## 5. Conclusions

A direct marketing strategy has been seen as an innovative way to improve the income of fishery farms. This study contributes to this research topic by examining its effect on aquaculture farms' fishery revenues, profits, and inputs used in aquaculture production between the use of traditional marketing channels and direct marketing. We drew a unique population-based dataset of aquaculture farm households from the census

survey in Taiwan. To quantify the effect of the use of marketing channels on the economic outcomes of aquaculture farms, we estimated a simultaneous equation system model with three choices of marketing channels and one outcome variable. After controlling for the socio-demographic characteristics of the farm operator, household and production condition, we found a significant and positive effect between direct marketing and fishery revenue and profit. Moreover, we found that revenue and profit are higher for farms that engage in multiple marketing channels. In addition, aquaculture farms engaging in direct marketing are less likely to use groundwater as the main water source in production.

Although this paper reveals several interesting findings, some caution is indicated. Perhaps one of the notable limitations is the use of the 2015 dataset. As indicated clearly in the paper, the census survey was conducted every five years, and the latest version is in 2015. Moreover, the 2015 census is the only available dataset that documents the use of marketing channels of fishery farms. Since the outbreak of COVID-19 that occurred in 2021 has disrupted the whole world, it may have affected the fishery industry as well. For example, it has been found that consumers' demand for online food shopping [29] and transportation and production costs of fishery products increased during the pandemic period [30]. This may increase the use of the marketing channels of fishery farms. Although we cannot obtain updated data to empirically accommodate the effect of COVID-19, we believe the main findings of this study can still stand during the COVID-19 period. The most significant result of this study indicates that the use of multiple marketing channels enhances fishery farm income in that using more than one channel to sell fishery products can help to spread the business operational risks of the farm. Given that transportation costs and market price became more volatile after COVID-19, the use of multiple channels may become more important to fishery farms to cope with these risks. Using the data after COVID-19 may strengthen our findings. This research topic can be better examined by future studies when the historical data are available in other countries or areas.

In addition to the issue of COVID-19, other caveats may remain. For example, in accordance with the information documented in the census survey, we can only define a binary variable for the use of direct marketing. If the data on product quantity sold to each marketing channel were available, we could measure the effect of the intensive margin of each marketing channel on fishery revenue. Moreover, direct marketing can be performed in several ways, such as online and offline sales. If more detailed data are available, we could further distinguish the impact on fishery revenue by different forms of direct marketing. Due to the limitation of data availability, we only know whether or not the farm operator worked in non-fishery work. If the information on the type of off-farm work was available, we could measure different forms or intensities of social networking. Finally, the amount of water used in fish production for each aquaculture farm was not documented in our data. This type of information could provide better insights into the link between water use and land subsidence. Regardless of these potential drawbacks, this paper is one of the first to provide an analytical framework and a case study to highlight the importance of marketing channels on fishery revenue and environmental quality.

**Author Contributions:** Conceptualization, T.-H.L., S.-Y.L., H.-H.C. and J.-H.W.; methodology, T.-H.L., S.-Y.L. and C.-L.H.; software, S.-Y.L.; validation, H.-H.C. and J.-H.W.; formal analysis, S.-Y.L. and C.-L.H.; writing—original draft preparation, T.-H.L., S.-Y.L. and C.-L.H.; writing—review and editing, H.-H.C. and J.-H.W. All authors have read and agreed to the published version of the manuscript.

**Funding:** This research received no external funding.

**Institutional Review Board Statement:** Not applicable.

**Data Availability Statement:** The individual data of the census survey can be accessed with permission from the Council of Agriculture in Taiwan.

**Conflicts of Interest:** The authors declare no conflict of interest.

# Appendix A

**Table A1.** Estimation results of the simultaneous equation system (the outcome variable is fishery revenue).

| Variable | Wholesaler Markets Coef. | S.E | Wholesalers Coef. | S.E | Direct Marketing Coef. | S.E | Fishery Revenue Coef. | S.E |
|---|---|---|---|---|---|---|---|---|
| Wholesaler markets | | | | | | | 1.582 *** | 0.515 |
| Wholesalers | | | | | | | 1.322 *** | 0.113 |
| Direct marketing | | | | | | | 1.075 | 0.519 |
| Age_3039 | 0.054 | 0.173 | −0.046 | 0.162 | 0.033 | 0.180 | 3.120 | 2.757 |
| Age_4049 | 0.111 | 0.166 | −0.086 | 0.156 | 0.227 | 0.173 | 1.881 | 2.649 |
| Age_5059 | 0.156 | 0.165 | −0.123 | 0.155 | 0.298 | 0.172 | 2.910 | 2.633 |
| Age_6069 | 0.160 | 0.165 | −0.150 | 0.156 | 0.306 | 0.173 | 3.479 | 2.647 |
| Age_70 | 0.110 | 0.167 | −0.158 | 0.157 | 0.250 | 0.174 | 1.495 | 2.674 |
| Elementary | 0.007 | 0.050 | 0.154 *** | 0.043 | 0.056 | 0.046 | 2.190 *** | 0.778 |
| Junior high | 0.098 | 0.056 | 0.101 ** | 0.050 | −0.025 | 0.053 | 1.508 | 0.908 |
| Senior high | 0.069 | 0.058 | 0.088 | 0.051 | 0.008 | 0.055 | 3.708 *** | 0.933 |
| College | 0.052 | 0.064 | −0.076 | 0.057 | 0.066 | 0.034 | 7.764 *** | 1.063 |
| Male | 0.004 | 0.035 | 0.018 | 0.033 | 0.077 ** | 0.035 | 0.889 | 0.598 |
| HHSIZE_male | 0.016 | 0.012 | −0.024 ** | 0.011 | 0.003 | 0.012 | 1.120 *** | 0.206 |
| HHSIZE_female | 0.023 ** | 0.010 | −0.038 *** | 0.010 | 0.005 | 0.010 | 0.774 *** | 0.184 |
| Ratio_adult | 0.086 | 0.098 | 0.128 | 0.091 | −0.119 | 0.095 | 2.012 | 1.670 |
| Aqua_grouper | 0.087 | 0.048 | 0.334 *** | 0.050 | −0.250 *** | 0.052 | 14.063 *** | 0.911 |
| Aqua_milkfish | 0.026 | 0.040 | 0.548 *** | 0.038 | −0.222 *** | 0.039 | −11.254 *** | 0.686 |
| Aqua_tilapia | 0.056 | 0.041 | −0.332 *** | 0.033 | −0.069 | 0.038 | −13.831 *** | 0.698 |
| Aqua_shrip | −0.096 ** | 0.042 | 0.433 *** | 0.041 | −0.160 *** | 0.042 | −7.782 *** | 0.728 |
| Aqua_oyster | −0.307 ** | 0.146 | 0.793 *** | 0.121 | −0.258 ** | 0.113 | −22.142 *** | 2.414 |
| Aqua_clam | −0.355 *** | 0.059 | 0.576 *** | 0.046 | −0.687 *** | 0.051 | −10.638 *** | 0.834 |
| Type_marine | 0.060 | 0.135 | −0.245 ** | 0.116 | 0.684 *** | 0.112 | 7.097 *** | 2.338 |
| Type_brackish water | −0.007 | 0.029 | −0.020 | 0.028 | 0.251 *** | 0.030 | −1.010 ** | 0.509 |
| City | 0.583 *** | 0.036 | −0.490 *** | 0.032 | 0.561 *** | 0.033 | −1.533 *** | 0.520 |
| Mkt_employee | 0.016 *** | 0.001 | −0.003 *** | 0.001 | −0.005 *** | 0.001 | | |
| Mkt_land | 0.287 *** | 0.020 | −0.224 *** | 0.020 | −0.301 *** | 0.021 | | |
| Mkt_equip | 0.005 ** | 0.002 | −0.006 *** | 0.002 | −0.000 | 0.002 | | |
| Constant | −2.057 *** | 0.205 | 0.787 *** | 0.191 | −1.311 *** | 0.209 | 0.098 | 3.440 |
| $\sigma$ | | | | | | | 30.649 | 16.844 |
| $\rho_{12}$ | | | | | | | −0.378 ** | 0.150 |
| $\rho_{13}$ | | | | | | | 0.227 *** | 0.017 |
| $\rho_{14}$ | | | | | | | −0.118 | 0.089 |
| $\rho_{23}$ | | | | | | | −0.338 ** | 0.153 |
| $\rho_{24}$ | | | | | | | −0.020 | 0.021 |
| $\rho_{34}$ | | | | | | | 0.028 | 0.020 |
| Log-likelihood | −144,948 | | | | | | | |

Note: *** and ** indicate significance at the 1% and 5% level.

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
