# Peer review of "Can Direct Marketing Increase Fishery Profitability and Environmental Quality? Empirical Evidence of Aquaculture Farm Households in Taiwan"

_agriculture, doi:10.3390/agriculture13061270_

Round 1

Reviewer 1 Report

Dear Authors

I have suggested a few points to strengthen the paper.

 The abstract seems fine. However, cut down the size and be specific. Check guidelines too. Avoid using the paper, the study is fine.

The introduction is too lengthy and the first few paragraphs can be written in a single paragraph. The introduction should be in 2-4 paragraphs: background, rationale, RQs and section plan. Please highlight RQs. Provide a section plan. Cut down the size and be crisp and clear.

Two things are missing: The operational definition of constructs and the supporting literature.

The data seems okay.

Results: There is nothing called 10% significance. If you check normal distribution, 1% has non-coverage of approx. 1 %, 5% has 5%, but 10% has 32% non-coverage.RECHECK.

Discussion: Must be discussed in comparison with past findings. No literary comparison is available, as you missed the operational definitions and literature review. ENHANCE.

Implications are missing. Both theoretical and practical (MUST) must be provided in any study.

Please REORGANIZE as per the suggestions.

All the best.

It seems okay.

Reviewer 2 Report

This is an interesting paper, which gives a general view on the effect of adoption of wholesaler markets, individual wholesalers or shippers, and direct marketing among aquaculture households in Taiwan. Although it relies on secondary data, the paper manages to find some interesting effects that manage to spring an interesting discussion. I believe that the last part of the analysis is slightly forced and the effects observed can be explained by various other variables that have not been taken into account. On the hole I believe this is an interesting paper and it raises interesting points and avenues for further research

Some rephrasing is also recommend, as the paper shows that English is not the native language of the authors. 

Reviewer 3 Report

1. Overall Comment: This study is highly interesting and consists of a very large sample. The introduction part offer insights on the marketing & distribution practice of aquaculture farm households. The contribution statement is also clear and appropriate. However, I believe its still presents a number of important issues that need to be necessary addressed in the attempt to make it publishable.

2. Some statements (lines 113-127) could be relocated to the Methods and Findings section. Rarely do researchers reveal the study's results in the introduction part.

3. The authors surveyed 25,180 aquaculture farm households. How much percentage is this from the overall aquaculture farms (non-family owned) operated in Taiwan?

4. On line 266 you wrote "These variables are drawn from the Statistics Yearbook of fishery production in 2015". Is this the latest publication? Considering its 8 years old, can you justify if it's relevant to reflect the current scenario?

5. The effect of a single direct marketing approach is the least feasible (scored the lowest coefficient) in predicting profit & revenue. You may discuss this implication.

6. On line 523-525, you wrote "our results provide an interesting case study that direct marketing can prove a win-win strategy to secure fishery revenue; it also has the potential to improve environmental quality." Perhaps you can justify, why the use of direct marketing allows farmers to avoid excessive groundwater pumping.

7. Similarly, based on Table 4, there are small adverse effects of the use of omni-marketing channels on groundwater use (positive beta coefficients). This finding could be explained as well in the discussion section.

Overall, it is a pleasant read. Good luck in this stream of research.

The use of english is OK.

Reviewer 4 Report

Dear Authors, 

the Data you used are really outdated (2015 survey data), since then the whole situation has significantly changed in particular due to climate change, the COVID-19 pandemic, etc. For this reason, the results are not at all covering the actual situation in the fish market. Just an example: In L277 you write: “Among the 25,180 aquaculture farm households, we found that 18,677 aquaculture farms selling fish products to wholesalers or shippers only (74%),” I really doubt that, the COVID-19 pandemic had a huge influence on marketing channels, direct to consumer sales increased, I am sure also in the fish market.

A further shortcoming concerns the structure of the manuscript, it is not adequate:

From L 113: The description of the data set should be part of the Materials and Methods section. 
L122: These are results and it is really surprising to find them already in the introduction section.
Also the whole paragraph starting in L129 with “This study contributes to previous studies on marketing channels in several ways. Firstly,…” should be put in another section (Discussion).
Chapter 2 should be part of the introduction, etc.
L188, Data – should be the chapter Materials and Methods 

ANOVA in L271 delivers rather results, aren’t they?

I suggest to follow the instructions of the Journal concerning the structure of the manuscript and adapt the text accordingly.

The econometric model seems to be correct (although I personally don’t appreciate 10% levels of significance) and well elaborated. In order to show that the data can be used to approximate marginal effects of the use of marketing channels, this would be ok. However, as you want to show the effects for your sample, the results are IMO not more than a historical analysis and are not really valid under actual market conditions (the world has dramatically changed during the last 8 years). You did estimate marginal effects of the use of marketing channels for the year 2015, but not for 2023. Todays’ fishery revenues are probably not based on the variables of the applied model. This is in particular relevant as you conclude in your discussion chapter: “We find that fishery revenues or profits are higher for aquaculture farms that involve multiple marketing channels.“. Is this still valid? At least, it is doubtable. And if so, your recommendations “The government should provide assistance or subsidies to increase the use of multiple marketing channels” would be wrong under actual conditions. You did not mention these limitations in your contribution, but even then I would suggest to resubmit the paper under a new general goal: to prove that it is possible to apply your econometric model for the purpose to explain fishery income and profits and the application of marketing channels.

Altogether, I cannot really suggest to accept the manuscript for publication (as it is now) mainly due to the outdated data set you used for your analysis. In my opinion, it could be beneficial to show that the model is capable to explain, e.g., the use of multichannel marketing and the potential benefits (by applying historical data from 2015). But the actual purpose going towards conclusions for the actual market behavior is at least doubtable if not flawed.

Round 2

Reviewer 1 Report

Hi

You almost answered everything. Thanks for accepting my suggestions

All the best.

Seems okay

Reviewer 4 Report

Dear Authors,

I still have concerns about the 2015 data set, but at least you now discuss this limitation in your discussion section. Therefore, I changed my opinion. As you also included all other issues in the revised version of your manuscript, I will suggest to accept the paper.

All the best for your future research.